# Ecofriendly Preparation and Characterization of a Cassava Starch/Polybutylene Adipate Terephthalate Film

**Tan Yi [1]** , **Minghui Qi [1]**, **Qi Mo [1]**, **Lijie Huang [1],\***, **Hanyu Zhao [1]**, **Di Liu [2]**, **Hao Xu [1]**, **Chongxing Huang [1]**, **Shuangfei Wang [1]** and **Yang Liu [1]**

[1]  College of Light Industry and Food Engineering, Guangxi University, Nanning 530004, China;
   YITANgxu@163.com (T.Y.); mhqi9812@163.com (M.Q.); MoQiGX@163.com (Q.M.);
   hannaxi08@163.com (H.Z.); liberxu@163.com (H.X.); huangcx21@163.com (C.H.);
   wshuangfei@163.com (S.W.); xiaobai@gxu.edu.cn (Y.L.)

[2]  Guangxi Key Laboratory of Clean Pulp & Papermaking and Pollution Control, Nanning 530004, China;
   ld1758352462@163.com

\*  Correspondence: jiely165@gxu.edu.cn

**Abstract:** Composite films of polybutylene adipate terephthalate (PBAT) were prepared by adding thermoplastic starch (TPS) (TPS/PBAT) and nano-zinc oxide (nano-ZnO) (TPS/PBAT/nano-ZnO). The changes of surface morphology, thermal properties, crystal types and functional groups of starch during plasticization were analyzed by scanning electron microscopy, synchronous thermal analysis, X-ray diffraction, infrared spectrometry, mechanical property tests, and contact Angle and transmittance tests. The relationship between the addition of TPS and the tensile strength, transmittance, contact angle, water absorption, and water vapor barrier of the composite film, and the influence of nano-ZnO on the mechanical properties and contact angle of the 10% TPS/PBAT composite film. Experimental results show that, after plasticizing, the crystalline form of starch changed from A-type to V-type, the functional group changed and the lipophilicity increased; the increase of TPS content, the light transmittance and mechanical properties of the composite membrane decreased, while the water vapor transmittance and water absorption increased. The mechanical properties of the composite can be significantly improved by adding nano-ZnO at a lower concentration (optimum content is 1 wt%).

**Keywords:** TPS; PBAT; nano-zinc oxide; plasticization; mechanical properties

## 1. Introduction

Plastic products have the advantages of light weight, beneficial mechanical properties, and excellent water resistance; however, their use is a double-edged sword. Some refractory high-molecular weight polymers have caused severe "white pollution" problems after disposal, and thus, the issue concerning the lack of petroleum resources needs to be solved. The use of biodegradable materials is one possible effective strategy for resolving these two problems. The application of biodegradable packaging has gradually become a trend. In addition, researchers have used low-cost natural degradable materials for blending with biodegradable polymers to prepare composite materials and achieve certain results [1–4].

Polybutylene adipate terephthalate (PBAT) is a ternary copolyester synthesized from adipic acid, terephthalic acid, and 1,4-butanediol through direct esterification or ester exchange polymerization. PBAT has good biodegradability and is one of the most widely used degradable polymer materials currently in use. It has been widely used in various fields such as daily plastic bags, food packaging

bags, garbage bags, agricultural mulch films, greenhouse insulation films, and other fields [5–7]. However, due to the presence of ester groups, the degradation rate of PBAT is very slow and its price is relatively high, which limits its wide application in the market [8]. At present, starch is regarded as the best additive to reduce the price of PBAT and improve its degradability [9]. Mixing starch with PBAT can not only improve the mechanical properties and reduce material cost, but also improve the degradation performance of the material and broaden the application range of PBAT, which is of great research significance [10]. Starch, which is the second largest class of polysaccharides in nature, is a renewable, abundant, and easy thermoplastic processing natural biodegradable material that is easily decomposed by microorganisms in nature [11]. However, a pure starch film is fragile and has poor mechanical and barrier properties, which lead to its limited application in degradable materials, packaging, medicine, food, and other industries. Thermoplastic starch (TPS) is obtained from native granular starch in the presence of a plasticizer [12], which can cause the surface groups of original starch to be replaced, thereby reducing the hydrophilicity of starch and improving its compatibility with petroleum materials; so normally, thermoplastic starch was used instead of native starch.

Many studies have been conducted on the potential advantages of blending TPS with poly (PBAT) [13–15]. Due to the difference in surface polarity between TPS and PBAT, the properties of TPS/PBAT composite materials largely depend on their compatibility. At present, major studies have adopted the method of adding a compatibilizer to improve the compatibility of starch with PBAT, such as the use of organic acid through an esterification reaction to replace starch surface hydroxyl, increase starch lipophilic and improve material properties [16–18]. Li Ming et al. used maleic anhydride, glycidyl ether, and chain extender (Joncryl-ADR-4368) to greatly improve the properties of TPS/PBAT composites and reported that 60% blending of PBAT improves TPS tensile strength by ~20% and elongation at break by more than 500% [18]. Ren et al.'s results suggest that addition of solvent and anhydride greatly improved the compatibility between starch and PBAT blend interface, and greatly improved the mechanical properties of the TPS/PBAT composite [19]. Olivato et al. explored the effects of the addition of citric acid and tartaric acid on the TPS/PBAT composite material and found that the addition of low content of organic acid could effectively increase the compatibility of TPS and PBAT and improve the uniformity of the membrane [6,16,20]. Meanwhile, blend ratio was the primary determinant of crystalline peak intensity for both TPS and polyester, and influences blend morphology [14,21]. Van Soest and Knooren found that $V_h$- and B-type crystallinity increased in potato starch TPS sheets and that it can strengthen and harden materials [22] In addition, it has been found that the addition of nanoparticles can increase the compatibility of materials and improve the mechanical properties of composites. Seligra et al. found that the addition of starch nanoparticles can increase the degree of gelatinization of starch, improve the compatibility of plasticized starch and PBAT, and enhance the mechanical properties of composites [23]. Feiya Fu and Lingyan Li et al. found that nano zinc oxide can improve the mechanical properties of biomass materials such as cellulose. After adding nano zinc oxide with a hexagonal shape and an average size of 18.6 nm, the tensile strength of the cellulose film showed a 28.9% increase in strength and 19.4% increase in elastic modulus [24]. Lendvai et al. modified TPS/PBAT composites with bentonite and modified montmorillonite to improve their ductility and mechanical properties in the glass state [25]. Meanwhile, previous research revealed that nanoparticles were very promising for improving the comprehensive performances of PBAT/TPS composites.

Our research aims at studying the effect of cassava starch after plasticizing with glycerin and organic acids, investigating the optimal blending ratio of plasticized starch and PBAT, exploring the effect of nano-zinc oxide on the mechanical properties of TPS/PBAT, and determining the optimum amount of nano-zinc oxide of the composites. Herein, in this study, properties of the cassava starch have been modified by reacting with the citric acid, lauric acid, and glycerol. After making plasticized starch, the material was combined with PBAT, using polyethylene glycol (200) as a compatibilizer to improve the interface between TPS and PBAT. Compatibly, TPS/PBAT composites were prepared, and the composites were characterized. Then, the polyethylene glycol (200) was used as a compatibilizer and

nano-zinc oxide particles were added to the composite for the purpose of investigating the mechanical properties of TPS/PBAT/nano-ZnO composites.

## 2. Methods and Materials

### 2.1. Materials

The main materials were cassava starch (food grade, Guangxi Wuzhou Triangle Food Co., Ltd., Wuzhou, China); glycerol, citric acid, dichloromethane, and potassium bromide (analytical grade, Tianjin Zhiyuan Chemical Reagent Co., Ltd., Tianjin, China); lauric acid (analytical grade, Tianjin Damao Chemical Reagent Co., Ltd., Tianjin, China); PBAT (Jining Hengtai Chemical Co., Ltd., Jining, China); and nano-ZnO (prepared in our laboratory).

### 2.2. Main Instruments

Thermostatic magnetic stirrer, type 85-2, Changzhou Putian Instrument Manufacturing Co., Ltd.; DC booster electric mixer JJ-1 type, Jintan Medical Instrument Factory.

### 2.3. Preparation of Materials

#### 2.3.1. Preparation of Thermoplastic Starch (TPS)

First, 10 g of cassava starch and 100 mL of distilled water were placed in a beaker. Then, lauric acid and citric acid (2% and 3% by mass, respectively, based on the quantity of starch) were dissolved in absolute ethanol and mixed with the starch. Next, 30% glycerin (with respect to the amount of starch) was added into the starch emulsion. The starch emulsion was mixed at room temperature by magnetic stirring, and then placed in a 95 °C constant-temperature water bath for 30 min to completely gelatinize the cassava starch. After cooling, the cassava starch was dried in a drying oven (DGG-9053AF, Shanghai Senxin Experimental Instrument Co., Ltd., Shanghai, China) at 80 °C for 48 h, pulverized in a pulverizer, passed through a 200-mesh sieve, and stored in a desiccator.

#### 2.3.2. Preparation of PBAT/TPS Composite Films

Plasticized starch was weighed (0.5, 1, 1.5, and 2 g) and added to 100 mL of dichloromethane. Subsequently, a corresponding mass of PBAT was added, and the total amount of cassava starch and PBAT was kept constant at 10 g. Polyethylene glycol 200 (1% by mass) was then added and the mixture was magnetically stirred for 2 h. After the PBAT was completely dissolved, the film-forming solution was poured onto a film-coating machine to form a film, and after dichloromethane completely volatilized, the film was peeled off.

#### 2.3.3. Preparation of PBAT/TPS/Nano-ZnO Composite Films

Nano-ZnO was weighed (0.05, 0.1, 0.15, and 0.2 g) and added to 100 mL of dichloromethane. Plasticized starch (1 g), PBAT (9 g) and Polyethylene glycol 200 (1% by mass) was then added and the mixture was magnetically stirred for 2 h. After the PBAT was completely dissolved, the film-forming solution was poured onto a film-coating machine to form a film, and after dichloromethane completely volatilized, the film was peeled off.

### 2.4. Characterization

#### 2.4.1. Scanning Electron Microscopy (SEM)

The film was placed in liquid nitrogen to make it brittle, and then fractured. The starch or film specimen was sputter-coated with gold using an ion sputtering instrument (SCB-12, Shanghai Minyi Electronics Co., Ltd., Shanghai, China) and fixed on a sample stage using a conductive adhesive. The microscopic morphologies of the samples were observed by SEM (Desktop scanning electron

microscope, Phenom World Pro, FEI, Hillsboro, OR, USA), and the changes in starch morphology and the association of PBAT with starch were analyzed.

### 2.4.2. Fourier Transform Infrared (FTIR) Spectroscopy

The dried powder sample (10 mg) was mixed with 300 mg of potassium bromide and formed into a pellet (YP-2 tablet press, Shanghai Shanyue Scientific Instrument Co., Ltd., Shanghai, China). Then, the powder or film sample (cut into a 1 cm × 1 cm piece) was scanned using a FTIR spectrometer (Bruker Corporation, Karlsruhe, Germany) in the scanning range of 4000–400 cm$^{-1}$ for chemical identification by observing the characteristic infrared absorption peaks.

### 2.4.3. Determination of Mechanical Properties of Films

According to the GB/T 1040.1-2006 standard, the obtained PBAT films were cut into rectangular strips having a length of 70 mm, width of 10 mm, and thickness of 0.09 mm, and tested using an electronic universal material testing 3376 (Instron, BSN, MA, USA) at a stretching speed of 50 mm/min [24,26]. Each group of samples was tested five times. The obtained data was calculated using Origin to calculate the homogeneity, standard deviation, and the confidence interval was calculated with a 95% confidence level.

### 2.4.4. Determination of Water Vapor Transmission (WVT)

The water vapor barrier properties of the film were tested by sampling with a standard sampler at a relative humidity of 90% and temperature of 38 °C according to GB/T 16928 and GB/T 1037-1988 standards [27,28]. Each test group was repeated five times and averaged.

### 2.4.5. Determination of Water Absorption

According to the national standard GB/T 1034-2008, the sample was cut into a square strip of 5 cm × 5 cm, dried to a constant weight at 105 °C, and weighed. Then, the sample was placed in a sealed bottle containing distilled water and immersed at room temperature for 72 h [29]. Afterward, the film was taken out from distilled water, the water adhering to the film surface was wiped dry, and the film was weighed. The water absorption rate was calculated according to the following formula, and five sample groups were measured and the results averaged.

Water absorption rate:

$$\omega(\%) = \frac{m_2 - m_1}{m_1} \times 100\%$$

Here, $m_1$ is the weight of sample before water absorption, and $m_2$ is the weight of sample after water absorption.

### 2.4.6. Determination of Contact Angle

The sample was cut to the size of the sample stage and placed on the sample stage of a contact angle goniometer (Lux, DSA100, KRUSS Corporation, Hamburg, Germany). Then, the sample stage of the contact angle meter, droplet size, and rate were adjusted. Water droplets were dropped on the thin film surface, and each set of samples was measured ten times and the results averaged.

### 2.4.7. Determination of Light Transmittance

A smooth and intact sample was selected, cut into a size of 10 mm × 30 mm, and attached to the surface of a cuvette. The light transmittance of the sample was measured at 650 nm using a UV–Vis spectrophotometer (SPECORD Plus 50, Jena Analytical Instrument Co., Ltd., Jena, Germany), and five sample groups were measured and the results averaged.

### 2.4.8. Thermal Analysis

The thermal stabilities of pristine starch and TPS dried to a constant weight in an oven at 45 °C were tested using a synchronous thermal analyzer (STA449F5, NETZSCH Corporation, BY, Selb, Germany). Approximately 5–10 mg sample was heated in the temperature range of 25–550 °C at a heating rate of 10 °C/min in a nitrogen flow rate of 10 mL/min. The thermal analysis data were automatically collected by the instrument [10].

### 2.4.9. X-Ray Diffraction (XRD) Analysis

The crystallinity of the samples was measured using an X-ray diffractometer (MiniFlex600, Rigaku Corporation, TKY, Japan), and the crystal structure of cassava starch before and after plasticization was analyzed. The detection conditions were 40 kV voltage, 250 mA current, Cu-Ka radiation (λ = 1.542 Å), $2\theta$ scanning range of 360°, and scanning speed of 10°/min.

## 3. Results

### 3.1. Characterization of Plasticized Starch

### 3.1.1. Microstructure Analysis of Raw Starch and TPS

As can be seen from Figure 1, after milling and screening through 200 mesh, the size distribution of raw starch was 5~20 μm, and that of TPS was 20~80 μm, in which most of the natural cassava starch is smooth and round, oval, and granular. After plasticization, the starch surface becomes rough, the starch granules are destroyed, and the cassava starch no longer shows dispersed granules. In addition, the boundary between the starch granules has disappeared. One possible mechanism for this is that citric acid, a ternary acid containing a hydroxyl group, forms an acidic solution with water in the starch emulsion, destroying the crystalline structure of cassava starch. The other possible mechanism is that the breakage of cassava starch granules under a high temperature and mechanical shearing force by citric acid and lauric acid reduces the length of the molecular chain of starch to a certain extent, making the molecular structure loose. This facilitates the entry of glycerol into the cassava starch chain forming a more stable thermoplastic starch, and the starch granules converting into plate-like particles.

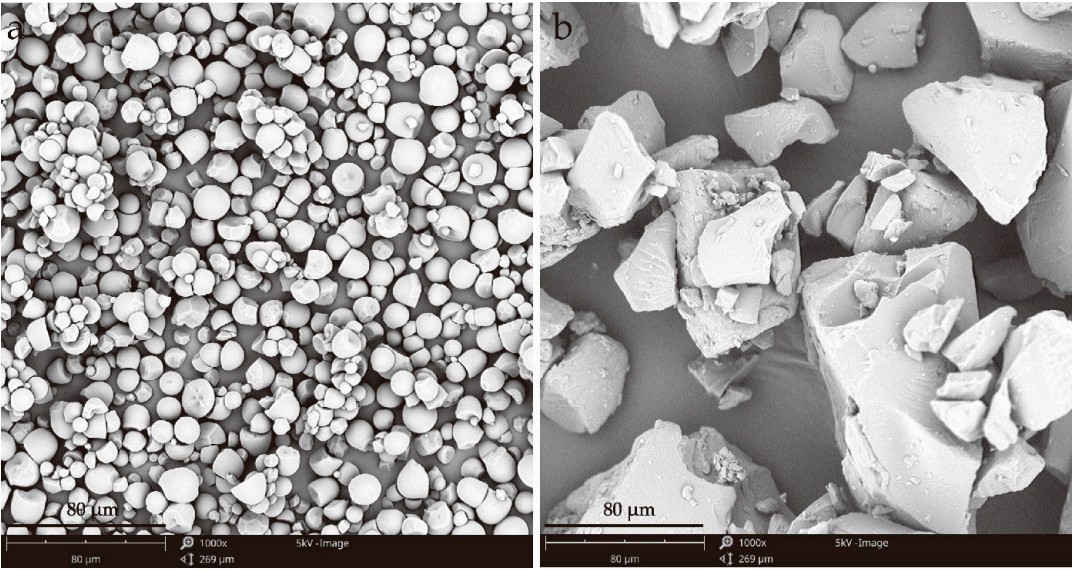

**Figure 1.** Scanning electron micrographs of (**a**) pristine starch (1000×) and (**b**) plasticized starch (1000×).

### 3.1.2. Thermal Analysis of Starch and TPS

Figure 2 shows that the thermal decomposition of cassava starch occurs in two stages: the first at 70~150 °C, which is mainly because of the evaporation of adsorbed and bound water [15]; and the second at 280~420 °C, which is mainly due to starch decomposition; sample quality drops rapidly, and the quality loss in this stage is greater [18]. At this stage, the weight loss rate of starch was 71.1%.

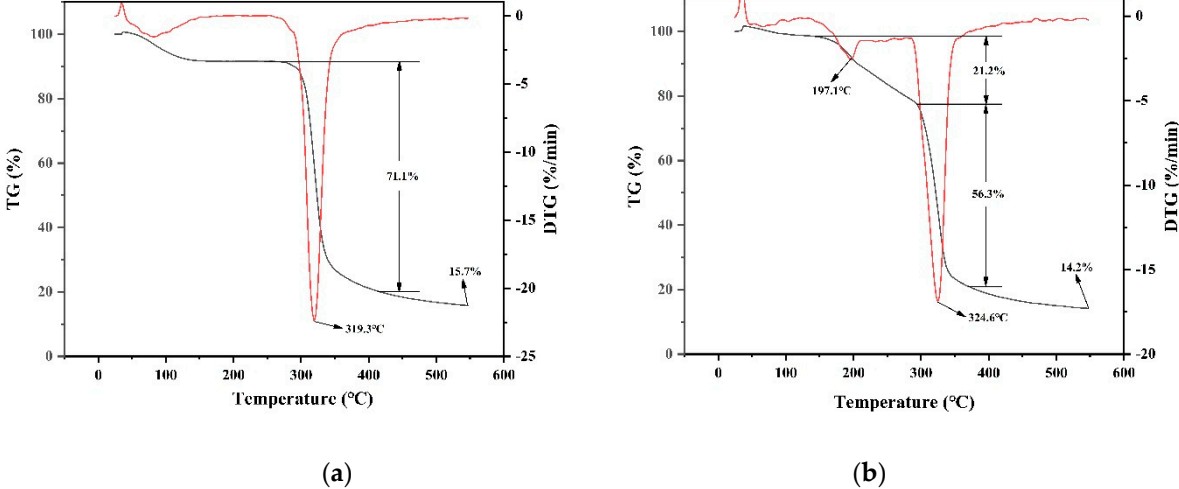

(a)　　　　　　　　　　　　　　　　　　(b)

**Figure 2.** TG and DTG curves of pristine starch (**a**) and plasticized starch (**b**).

The thermal decomposition of TPS occurred in three stages: the first at 50~120 °C, which is mainly because of the evaporation of adsorbed water; the second at 155~295 °C, which is due to the decomposition of glycerin and citric acid; the weight loss rate of TPS was 21.2%; and the third at 295~370 °C, which is mainly because of lauric acid and starch decomposition. In this stage, the weight loss rate is 56.3%. The initial weightlessness temperature of plasticized starch is lower than that of pristine starch due to the loss of the plasticizer, glycerol. In the final stage of thermal decomposition, the residual mass of TPS is lower than that of pristine starch. Therefore, the thermal stability of plasticized starch decreased slightly as compared with pristine starch.

### 3.1.3. XRD Analysis

As illustrated in Figure 3, the crystal structure and type of starch changed significantly before and after plasticization. Cassava starch shows three strong diffraction peaks at 2θ = 15.08°, 17.94°, and 22.9°, which indicate type A crystal structure. After plasticization, only one strong diffraction peak appeared at 2θ = 20.18°, indicating a change in crystal type [30]. In addition, compared to the original starch, the crystallinity of the plasticized starch also changed. According to jade analysis, the crystallinity of the original starch was 23.9%, and the crystallinity of the plasticized starch was 74.55%. The improvement in the crystallinity of plasticized starch is mainly due to the opening of starch molecular chains and the replacement of hydrogen bonds between starches by organic acids and glycerol.

Starch has a double helix structure of starch molecules with a large number of hydroxide groups forming intramolecular and intermolecular hydrogen bonds. When water molecules or glycerin form hydrogen bonds with the hydroxyl groups of starch, the intramolecular and intermolecular hydrogen bonding interaction in starch weakens, and the double helix structure of starch completely opens. Thus, the plasticizer molecules form new stable hydrogen bonds with starch molecules, altering the crystal structure of starch [15,31]. Plasticizing starch 2θ, 20.18° in representative V single spiral crystal diffraction peak [21]. The V-type structure is rarely found in natural starch. It is mainly formed by the complexation of amylose with fatty alcohols and fatty acids. In this study, the V-type crystal structure could have formed by a reaction between cassava starch and lauric acid [22].

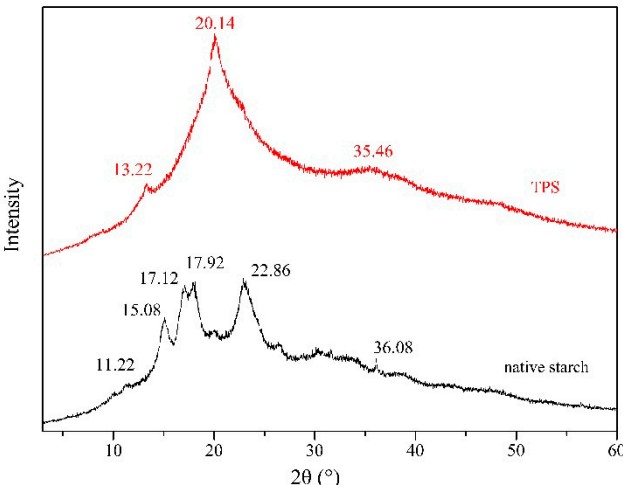

**Figure 3.** XRD spectra of pristine cassava starch and plasticized starch.

### 3.1.4. FTIR Spectroscopy Analysis

As Figure 4 illustrates, the FTIR spectrum of pristine cassava starch shows characteristic peaks at 3247, 2930, and 1643 cm$^{-1}$ which are the stretching vibration peak of O–H under molecular association, -CH$_2$ stretching vibration, and symmetric bending vibration of the C=C double bond, respectively. The absorption peaks at 1018, 1083, and 1158 cm$^{-1}$ can be ascribed to the C–O stretching vibration of primary alcohol, secondary alcohol and tertiary alcohol, respectively, where the vibration peak of primary alcohol represents the anti-symmetric stretching vibration of C–O–C. 579 cm$^{-1}$, 710 cm$^{-1}$, 859.38 cm$^{-1}$ is the out-of-plane bending vibration peak of C–H and O–H.

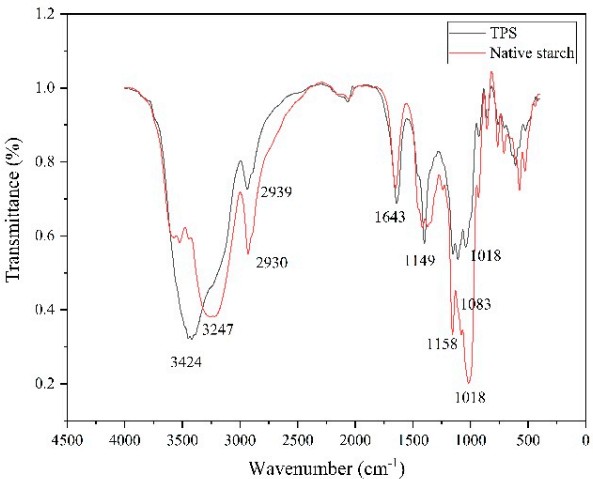

**Figure 4.** FTIR spectra of pristine cassava starch and plasticized starch.

In the FTIR spectrum of plasticized starch, the characteristic peak at 3424 cm$^{-1}$ is blue-shifted compared with the peak in the spectrum of pristine cassava starch. This is because organic acids and glycerol form hydrogen bonds with the hydroxyl groups on the starch, forming a new hydrogen bond structure, and reducing the intra- and intermolecular hydrogen bonding of the starch, which reduces the content of multi-association in cassava starch. Further, the absorption peak at 2939 cm$^{-1}$, which resulted from –CH$_2$ stretching vibration, weakened as compared with that of pristine starch. The C–O bond stretching vibration peaks of 1018cm$^{-1}$, 1102cm$^{-1}$ and 1149cm$^{-1}$ can be divided into primary, secondary and tertiary alcohols; 607 cm$^{-1}$, 757 cm$^{-1}$, 859 cm$^{-1}$ are the out-of-plane bending vibration peaks of C–H and O–H. The esterification of pristine starch with several functional groups of

an adjuvant changed the molecular structure of starch, yielding plasticized starch that gave rise to a peak at 3424 cm$^{-1}$. Moreover, the small vibration peaks nearly disappeared, and new small vibration peaks appeared at 1149–1018 cm$^{-1}$, formed by the grafting of lauric acid and citric acid onto the starch skeleton [22,32]. The addition of lauric acid to citric acid also improved the lipophilicity of TPS particles and improved the compatibility of TPS with PBAT in blending.

### 3.2. Characterization of Composite Film

#### 3.2.1. Microstructure Analysis of TPS/PBAT Composite Films

As can be seen from Figure 5, the fracture surface of the pure PBAT film is smooth and flat. Less starch was added to b and c. Plasticized starch is almost invisible in the section. However, with an increase in the amount of plasticized starch, distinct plasticized starch particles can be seen in the fracture surfaces of 15% and 20% TPS/PBAT (TPS/PBAT composites containing TPS mass fractions of 15% and 20%) composite films. In addition, with an increase in plasticized starch content, the surface of the flat film becomes increasingly rough, uneven, and undulating, with uneven film thickness and increase number of holes. The addition of plasticized starch destroys the integrity and continuity of the pure PBAT film, thus reducing its mechanical properties, which also partly explains the reduction in the mechanical properties of composite films due to the addition of plasticized starch. However, the introduction of hydrophilic starch particles can also make it easier for water molecules to attack PBAT and improve the biodegradability of composite films [33,34].

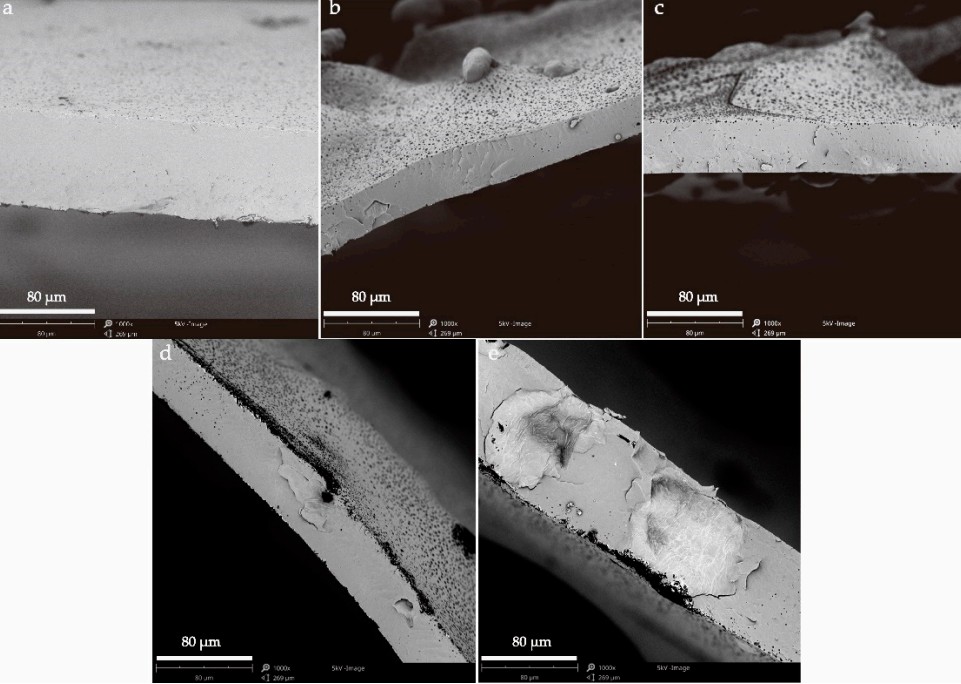

**Figure 5.** Scanning electron micrographs of film sections showing fracture surfaces: (**a**) pure PBAT film (1000×), (**b**) 5% TPS/PBAT composite film (1000×), (**c**) 10% TPS/PBAT composite film (1000×), (**d**) 15% TPS/PBAT composite film (1000×), and (**e**) 20% TPS/PBAT composite film (1000×).

#### 3.2.2. FTIR Spectroscopy Analysis

Figure 6 shows the FTIR spectra of the pure PBAT film and 10% plasticized starch/PBAT film. Compared with the pure PBAT film, the plasticized starch/PBAT film shows a broad peak at 3400 cm$^{-1}$, which is attributed to the stretching vibration of –OH due to the presence of a large number of hydroxyl groups in the plasticized starch. Therefore, the addition of plasticized starch should reduce the hydrophobicity of the film. The peak at 2953 cm$^{-1}$ can be ascribed to the stretching vibration of –CH$_3$

and –CH$_2$. The strong absorption peaks at 1720 and 1722 cm$^{-1}$ correspond to the carbonyl group in PBAT, whereas weak absorption peaks corresponding to –CH$_2$OH and –CH$_2$ appear at 1251 cm$^{-1}$. The asymmetric C–O–C stretching vibration and C–O stretching and skeleton vibration gave rise to an absorption peak at 723 cm$^{-1}$. The addition of plasticized starch to PBAT changed neither the position, nor the intensity of the characteristic infrared absorption peaks of the film. Therefore, the mixing of plasticized starch to PBAT was a purely physical process.

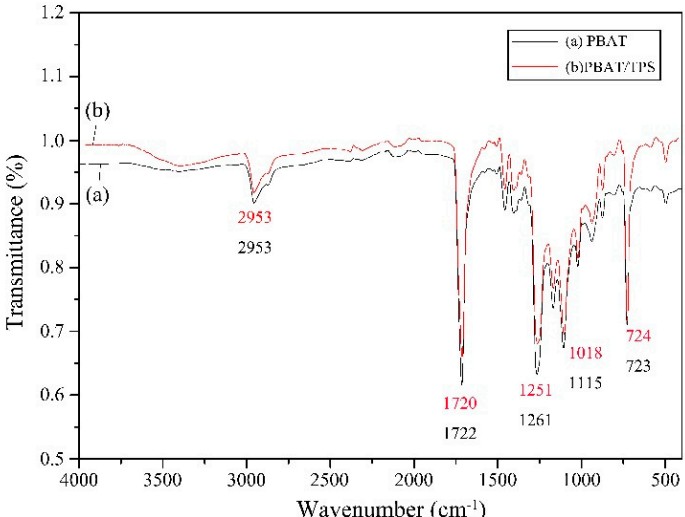

**Figure 6.** FTIR spectra of PBAT and 10% plasticized starch/PBAT.

### 3.2.3. Transmittance and WVT Analyses

Figure 7 shows that the light transmittance of pure PBAT film is 40.41%, because TPS particles are non-transparent materials and have a strong reflection and absorption effect on light. With an increase in the content of plasticized starch, the light transmittance of the composite decreased gradually. Compared with the pure PBAT film, the light transmittance of the PBAT film containing 5% plasticized starch decreased by 4.5%. When the amount of TPS is 10%, the transmittance of the composite material decreases by 14.2%. At a plasticized starch content of 15%, the light transmittance of the composite decreased by 23.2%. Therefore, the composite film can be used in agriculture as a low-light transmittance film, and as environmentally friendly shopping bags. Compared with the pure PBAT film, the composite film containing starch has the advantage of low price as well as complete biodegradability from the perspective of environmental protection [17,35].

It can be seen that the water vapor transmission rate of the pure PBAT film is quite high (108.13 g/m$^2$·24 h), and the WVT of the film is promoted with an increase in TPS content simultaneously. It can be ascribed that TPS introduces the alcohol hydroxyl group of the glycerol, reducing the hydrophobicity of the composite film [17]. This result is consistent with the results for the TPS/PBAT films. At the TPS content of 5%, the WVT of the film increase by 2.6%, and the effect on the barrier property of the composite film is small. The WVT of the composite film with 10% TPS increase by 14.6%, which that of the composite film with 15% TPS increased by 29.2%. According to data analysis, with TPS contents of 5% and 10%, there is no obvious change of the water vapor barrier of the film, which not only ensures the barrier of the raw material, but also reduces the cost of PBAT material. Due to the poor water barrier properties of the PBAT and TPS/PBAT, this composite material can only be used to make products with loose requirement for the high water barrier properties, such as plastic packaging bags and agricultural mulch [36].

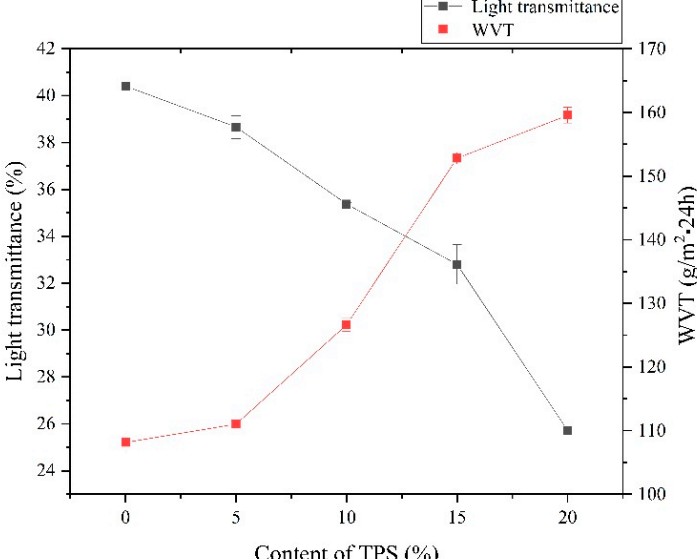

**Figure 7.** Effect of TPS content on the light transmittance and WVT of composite films.

### 3.2.4. Water Absorption Rate Analysis

As can be seen from Figure 8, the water absorption rate of the films with different starch contents in distilled water increased with an increase in TPS content over 72 h. This is because the surface of the plasticized starch contains a large number of hydrophilic groups, which easily absorbs moisture in the environment and produces a certain degree of swelling or melting and the low barrier of PBAT to water vapor also promotes this process. In addition, the water swelling of plasticized starch particles can also cause damage to the surface of composite materials making it easier for water to penetrate into the material and be absorbed by the TPS particles inside the material. Therefore, the increasing TPS content can lead to a more obvious penetration and increasing water absorption of the composite. In addition, the decrease of PBAT prevents PBAT from wrapping the TPS particles well, contributing to more TPS particles being exposed on the surface of the film which further increases the water absorption rate of the film.

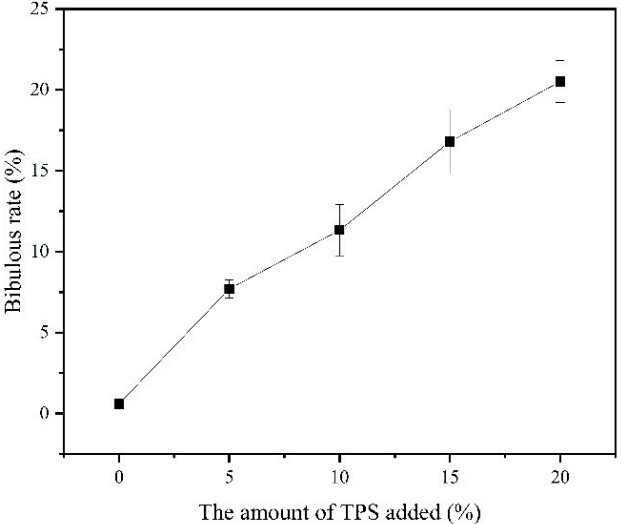

**Figure 8.** Effect of TPS content on the water absorption of composite films.

### 3.2.5. Contact Angle on TPS/PBAT Composite Films

Table 1 shows that PBAT has good hydrophobicity. However, since TPS is a hydrophilic organic compound, an increase in its content led to a decrease in the water contact angle of the films from 104.1° to 90°. Therefore, although the hydrophobicity of the film declined, it could still satisfy the requirements for packaging and general use and can be used for packaging and polluting plastics.

**Table 1.** Water contact angles of composite films.

| PBAT/TPS/Nano Zinc Oxide (w/w/w) | Contact Angle (°) |
|---|---|
| 100/0/0 | 104.1 ± 0.60 |
| 95/5/0 | 101.8 ± 0.38 |
| 90/10/0 | 93.6 ± 0.32 |
| 85/15/0 | 91.05 ± 1.71 |
| 80/20/0 | 90.0 ± 7.7 |
| 90/10/0 | 93.6 ± 0.32 |
| 0.5/90/10 | 91.05 ± 1.71 |
| 90/10/1 | 95.5 ± 0.18 |
| 90/10/1.5 | 92.2 ± 1.21 |
| 90/10/2 | 89.9 ± 1.30 |

Further, the contact angle data show that the addition of a low amount of nano-ZnO has little effect on the hydrophobicity of the material, and the films containing low amounts of starch and nano-ZnO remain hydrophobic.

### 3.2.6. Mechanical Properties of TPS/PBAT Composite Films

It can be seen from Figure 9 that once the plasticized starch is added, the tensile strength and the elongation at break of the composite material can be reduced dramatically. With the gradual increase in the amount of plasticized starch, the tensile strength of the composite film slowly decreases in the range of 5 to 15% of TPS, When the amount of TPS increases to 20%, the tensile strength decreases sharply. Compared with the tensile strength, the elongation at break decreased more significantly. As for the pristine PBAT film, the tensile strength and elongation at break are 16.84 MPa and 845%, respectively. As for the composite films, the tensile strength and elongation at break are 38.48% and 37.85% at the TPS content of 10%, and 65.75% and 45.55% at the TPS content of 20%, respectively. The elastic modulus of composite material is shown in Supplementary Material Figure S1. Since TPS particles and PBAT are physically composite, after adding TPS particles, the composite material is prone to generate large defects at the interface between the two, forming a stress concentration point to reduce the mechanical properties of the material. Thus, TPS will agglomerates when the TPS content increases, which could increase the probability to form larger agglomerates in the material and then lead to uneven distribution in PBAT; In addition, with the decreasing PBAT content, it is difficult for PBAT to infiltrate and wrap TPS particles effectively, easily causing the composite interface. Moreover, the interface bonding strength and the composite material strength will decrease due to existence of the cavity [34,36,37].

### 3.2.7. Mechanical Properties of TPS/PBAT/nano-ZnO Composite Films

Comprehensive consideration of water absorption, light transmission, water vapor barrier and mechanical properties of composite materials, the TPS/PBAT composite with TPS content of 10% is selected and nano zinc oxide is added to it to study the effect of nano zinc oxide on the tensile strength of TPS/PBAT composite; the results are shown in Figure 10. It can be seen from Figure 10 that the addition of nano-ZnO greatly improves the mechanical properties of the composite material. With the addition of nano-ZnO, the tensile strength and elongation at break of the composite material change first and then decrease, Where the tensile strength and elongation at break of the composite reached

their peaks when nano-ZnO was added at 1% and 0.5%, respectively. As can been seen, the tensile strength and elongation at break of the 10% TPS/PBAT composite film were 10.36 MPa and 525.24%, respectively. After the addition of 1% nano-ZnO, the tensile strength increased by 153.28% to 26.24 MPa and the elongation at break decreased by 3.16% to 541.81%; when 0.5% nano-ZnO was added, the elongation at break of the 10% TPS/PBAT composite increased by 15.27% to 619.89%. The elastic modulus diagram of composite material is shown in Supplementary Material Figure S2.

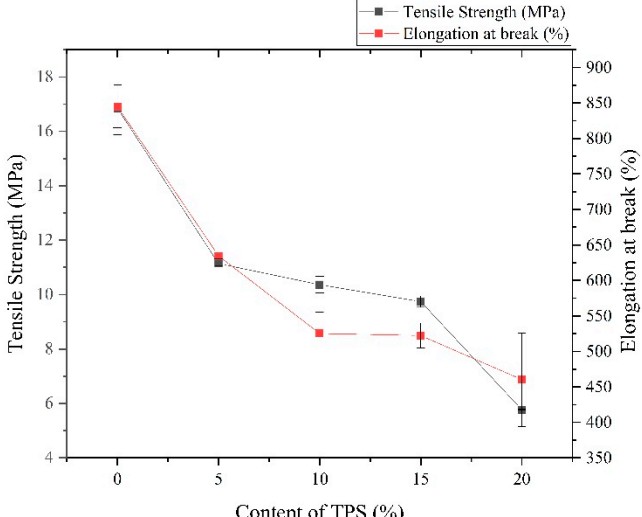

**Figure 9.** Effect of TPS content on the mechanical properties of composite films.

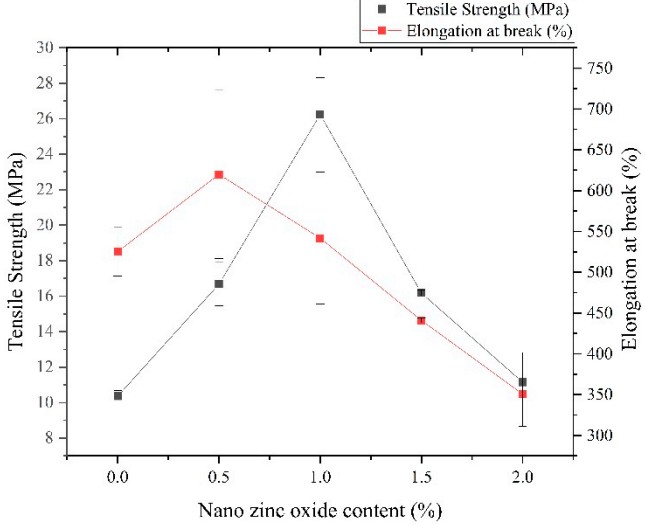

**Figure 10.** Effect of nano-ZnO content on the mechanical properties of composite films.

One of the possible reasons is that nano-ZnO particles fill the grooves on the surface of the slab-like plasticized starch particles, which reduces the roughness of the surface of the TPS particles. This also allows PBAT penetrating into the small depressions on the surface of the TPS particles through the tiny particles of nano-ZnO in the groove. The improvement of interface compatibility also improves the mechanical properties of composite materials. At the same time, the addition of nano-ZnO can hinder the large-scale movement of the PBAT segment to some extent during the stretching process, thereby further improving the tensile strength of the material. As the content of nano-ZnO increases, the agglomeration of nano-sized zinc oxide begins. Obviously, a tendency of its distribution in PBAT can be discovered to become non-uniform, resulting in a decrease in the mechanical properties of

the composite. In addition, the elongation at break of the composite material reaches its peak when the amount of nano-ZnO added to 0.5%, which can be attributed that the added extremely small amount of nano-ZnO agglomerates around the TPS particles under the effect of electrostatic attraction, preventing the composite material from pulling during the elongation process, the interface separation between TPS particles and PBAT occurs, which plays a certain lubricating effect and, thus, improves the elongation at break of the composite [38–40].

## 4. Conclusions

(1)   The pristine starch granules were spherical, whereas the TPS particles were plate-like. With an increase in TPS content, the cross section of the composite film became increasingly uneven, and the number of holes on the film surface increased. Compared with pristine starch, the lipophilicity increased, but the thermal stability of TPS decreased and its crystal structure and type changed.

(2)   In the case of plasticized starch, new small vibration peaks appeared in the range of 1149–1018 $cm^{-1}$ in the FTIR spectrum, which were attributed to the grafting of lauric acid and citric acid onto the starch skeleton. In addition, the characteristic peak of hydroxyl groups in plasticized starch moved to 3424 $cm^{-1}$. This is the result of organic acids and glycerol interacting with the hydroxyl groups contained in starch. The hydrogen bonds in starch are replaced by the new-forming hydrogen bonds, which effectively reduce the hydrogen bond content in starch. Besides, the addition of plasticized starch to PBAT changed neither the position nor the intensity of the characteristic infrared absorption peaks of the films. Therefore, it can be indicated from the results above that the mixing of plasticized starch to PBAT is a purely physical process.

(3)   The addition of TPS increased the water absorption of the composite film, and reduced the light transmittance, water vapor barrier, and average water contact angle. Moreover, the addition of nano-ZnO reduces the water contact angle of the composite film but does not affect its performance. However, the addition of nanometer zinc oxide increased the water contact angle of the composite film containing 10% TPS, and the effect is most obvious when the amount of nano-ZnO is 1%.

(4)   With an increase in TPS content, the tensile strength and elongation at break of the composite film decreased gradually, but compared with the tensile strength, the elongation at break decreased more. In addition, the addition of nano-ZnO can greatly improve the tensile strength of TPS/PBAT composites, and the addition of a certain amount of nano-ZnO can increase the elongation at break of the composite. For 10% TPS/PBAT composites, the addition of 1% nano-ZnO can increase the tensile strength of the composites by 15.88 MPa without substantially reducing the elongation at break of the materials. When nano-ZnO is added in an amount of 0.5%, the elongation at break of the composite can be increased by 15.27%.

**Supplementary Materials:** The following are available online at http://www.mdpi.com/2227-9717/8/3/329/s1, Figure S1: Elastic modulus of PBAT/TPS composite material, Figure S2: Elastic modulus of PBAT/TPS/ nano-ZnO composites, Table S1: Standard deviation and confidence interval of PBAT/TPS composite tensile strength data, Table S2: Standard deviation and confidence interval of elongation at break data for PBAT/TPS composites, Table S3: Standard deviation and confidence interval of tensile strength data of PBAT/TPS/ nano-ZnO composites, Table S4: Standard deviation and confidence interval of elongation at break data of PBAT/TPS/ nano-ZnO composites.

**Author Contributions:** T.Y., conceptualization, supervision and writing; T.Y., M.Q., H.Z. and D.L., conceptualization; H.X., Q.M., SEM and XRD investigations; L.H., C.H., S.W. and Y.L., project management; all authors, writing—review and editing. All authors have read and agreed to the published version of the manuscript.

**Funding:** The work was funded by Nanning Scientific Research and Technological Development Plan Project (20195215), Guangxi Key Laboratory of Clean Pulp & Papermaking and Pollution Control (No. ZR201806-6), Guangxi Science and Technology Project (Guangxi Science AB18221126) and the Construction Project Fund of the Guangxi Undergraduate College Specialty and Experimental Training Teaching Base (center) in 2018–2020, China (T3050094101).

**Conflicts of Interest:** The authors declare no conflict of interest. The sponsors had no role in the design, execution, interpretation, or writing of the study.

## Abbreviations

The following abbreviations are used in this manuscript:

XRD     X-ray diffraction
SEM     Scanning Electron Microscope
WVT     Water vapor transmission

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
