# Peer review of "Ecofriendly Preparation and Characterization of a Cassava Starch/Polybutylene Adipate Terephthalate Film"

_processes, doi:10.3390/pr8030329_

Round 1
Reviewer 1 Report
In this manuscript, Qi et al. have reported the fabrication of starch and polybutylene based composite films and characterized the morphology and physical properties using several characterization techniques. The study may be considered for publication once the following comments have been thoroughly taken care of:
- Scale bars in Figure 1 are not visible. I suggest authors to use the bigger text.
- Authors must include these relevant articles on the mechanical properties of biodegradable materials (ACS Nano 2018, 12, 7, 6378-6388; ACS Nano 2017, 11, 5, 5148-5159; Angew. Chem. Int. Ed. 2019, 58, 18562 – 18569).
- Figure 2 should be replaced with a clear figure.
- From the figure 3, authors should quantify the ratio of crystalline to amorphous regions.
- Scale bars in figure 5 need to be improved, it's not clearly visible.
- What's the elastic modulus of the films ?
- How many samples have been tested for the mechanical properties, how standard deviation has been calculated, whats the confidence interval ? Authors must report the relevant data.
Author Response
Dear reviewer:
The reply letter is submitted as an attachment.
Kind regards,
Tan Yi

Reviewer 2 Report
These authors have prepared filled Polyester through the use of Thermoplastic starch and nano-particle zinc oxide admixtures for use in the production of an eco-friendly degradable material. Their approach appears sound and the data and outcomes well supported by a whole range of methodogies to investigate the unusual interaction on the physical properties of the polymer. The articles in well-presented although I do feel in places some of the grammar is in need of editing.
On the scientific approach I am confused by the sudden choice of the nano zinc oxide other than it does enhance the physical properties of the polyester but only at low levels i.e. below 0.5-1.0 % w/w. They mention the formation of agglomerates by the oxide at high concentrations although it would have been useful to elaborate a bit more with some pictoral evidence. Their preparation of the dopant mixtures i.e. in dichloromethane leaves me a bit dubious as perhaps here the authors should have dispersed the mixtures thoroughly with a high speed mixer rather than a magnetic stirrer! The latter is very inefficient and will result in poorly dispersed mixtures. The agglomerates need to be broken up.
Author Response
Dear reviewer
The reply letter is submitted as an attachment.
Kind regards,
Tan Yi

Round 2
Reviewer 1 Report
Authors have considered the suggestions from reviewers. I believe this manuscript is now ready for publication in Processes.
Author Response
尊敬的审稿人:
感谢您的指导和对我们工作的肯定。
亲切的问候,
è°æ€¡
Reviewer 2 Report
No cover letter attached just a copy of the article when I click on authors-response
Author Response
Dear Reviewer,
I'm very sorry that I failed to upload the cover letter due to my work mistake.
Now the cover letter has been uploaded, please see the attachment.
kind regards,
Tan Yi

Round 3
Reviewer 2 Report
Thank you for the reply Ok I will go with it.